# Robust Optimization for Mitigating Reward Hacking with Correlated Proxies

## Abstract

Designing robust reinforcement learning (RL) agents in the presence of imperfect reward signals remains a core challenge. In practice, agents are often trained with proxy rewards that only approximate the true objective, leaving them vulnerable to reward hacking, where high proxy returns arise from unintended or exploitative behaviors. Recent work formalizes this issue using $r$-correlation between proxy and true rewards, but existing methods like occupancy-regularized policy optimization (ORPO) optimize against a fixed proxy and do not provide strong guarantees against broader classes of correlated proxies. In this work, we formulate reward hacking as a robust policy optimization problem over the space of all $r$-correlated proxy rewards. We derive a tractable max-min formulation, where the agent maximizes performance under the worst-case proxy consistent with the correlation constraint. We further show that when the reward is a linear function of known features, our approach can be adapted to incorporate this prior knowledge, yielding both improved policies and interpretable worst-case rewards. Experiments across several environments show that our algorithms consistently outperform ORPO in worst-case returns, and offer improved robustness and stability across different levels of proxy–true reward correlation. These results show that our approach provides both robustness and transparency in settings where reward design is inherently uncertain.

## 1 Introduction

Real-world reinforcement learning (RL) systems often struggle with reward specification: it is notoriously difficult to craft a reward function that perfectly captures the intended goals in all scenarios [1, 2, 3]. In practice, designers rely on proxy rewards that approximate the true objective [4]. However, agents optimizing these imperfect proxies can lead to unintended exploitative behaviors, achieving high proxy returns while yielding poor true outcomes, a phenomenon known as reward hacking [5, 6, 7, 8]. Such reward hacking behaviors are not merely hypothetical; they have led to undesirable or even catastrophic consequences in safety-critical settings (e.g., autonomous driving) [9, 10] and are alarmingly common in real-world deployments [11, 12, 13, 14]. Beyond reward hacking, interpretability and transparency of RL policies are increasingly recognized as critical requirements for real-world acceptance [15, 16, 17]. Policymakers and practitioners in safety-critical domains require systems not only to be robust but also interpretable; they must understand which specific decision-making criteria lead to undesirable outcomes to effectively mitigate risks and ensure compliance with safety regulations [18, 19, 20]. These challenges highlight the need for RL algorithms to address two fundamental challenges: robustness to uncertain or poorly-specified rewards, and interpretability to facilitate oversight and compliance by human stakeholders, especially in high-stakes, real-world environments like traffic control [21], healthcare decision-making [22, 23], and pandemic response strategies [24].

Submitted to 39th Conference on Neural Information Processing Systems (NeurIPS 2025). Do not distribute.

Recent work has begun to formalize reward hacking and develop principled mitigations. Laidlaw et al. [25] define a proxy reward to be *r-correlated* with the true reward if it maintains a correlation coefficient $r > 0$ on state-action pairs encountered by a certain reference (baseline) policy. Notably, their definition permits the proxy and true reward to diverge arbitrarily in parts of the state-action space not visited by the reference policy, precisely the regions an RL agent might exploit under intensive optimization. Using this framework, reward hacking is formalized as the situation in which optimizing an $r$-correlated proxy yields a policy with lower true reward than that of the reference policy. Building on this definition, Laidlaw et al. propose Occupancy-Regularized Policy Optimization (ORPO) as a mitigation strategy. ORPO augments the standard RL objective with a regularization term that penalizes deviations between the learned policy's occupancy measure (state-action visitation distribution) and that of the reference policy.

Despite significant progress, existing solutions to reward hacking show several limitations. First, their effectiveness relies heavily on the choice of the specific proxy reward. However, designing perfect proxies is challenging, and in real-world scenarios, reward proxies are often derived heuristically or empirically from noisy or limited data [26, 27], leading to uncertainty or variability in the exact correlation with true rewards. Therefore, robustness to variations in proxy rewards is crucial for dependable deployment. While the regularization method used by ORPO provides a lower bound on improvement in true reward, its guarantee on the worst-case performance against an adversarially chosen proxy is weak. Second, current methods like ORPO typically treat a reward function as a black box and learn a complex policy with no easily interpretable structure, making it hard to understand why the resulting policy avoids reward hacking or to trust its behavior in novel situations. Further, they cannot be easily adapted to incorporate prior knowledge of the true reward. These shortcomings underscore the need for a more robust and transparent approach to reward hacking in RL.

In this work, we formalize reward hacking as a robust RL problem under proxy reward uncertainty and develop new algorithms to address the above gaps. The key idea is to optimize against an adversarial proxy reward rather than trusting a single proxy. We assume the true reward could be any function that remains $r$-correlated with the proxy (per the reference policy), and we train the agent to perform well against the worst-case such proxy. This approach explicitly accounts for uncertainty in proxy design and guards against unintended exploitative behaviors. Concretely, we propose a max-min formulation in which the policy chooses its strategy to maximize its guaranteed true return while an adversary minimizes the true return by selecting a reward function from the set of all $r$-correlated proxies. By solving this problem, the agent learns a policy that is robust to all plausible deviations of the proxy reward within the correlation bound. We derive a closed-form solution for the adversary's worst-case reward assignment given any candidate policy, which allows efficient evaluation of the inner minimization and provides insight into how proxy reward flaws are most damaging. Building on this result, we introduce a practical algorithm for Max-Min Policy Optimization that iteratively updates the policy against this worst-case reward signal.

Moreover, to improve the tractability and transparency of the inner optimization, we introduce a Linear Max-Min variant of our method. In this variant, we assume the true reward lies in a class of linear functions over known features, allowing us to characterize the worst-case proxy reward as a sparse linear combination of those features. While the policy itself remains parameterized by general neural networks, the learned worst-case reward function becomes interpretable in terms of its feature weights. This provides insight into which aspects of the proxy reward space the policy is robust to or vulnerable against, making it valuable for applications where understanding the failure modes of the reward design is important.

Finally, we empirically evaluate the proposed approaches on several challenging environments. Across all domains, our Max-Min and Linear Max-Min policies outperform ORPO in terms of worst-case reward, indicating substantially improved robustness. Moreover, under a large range of proxy-true correlation scenarios, our methods exhibit higher average reward and lower variance compared to ORPO, meaning the performance of our policies remains more consistent and reliable. These findings demonstrate the practical significance of our robust formulation, paving the way for safer and more trustworthy RL deployment in real-world applications.

Our main contributions can be summarized as follows: 1) We propose a novel robust RL formulation that explicitly models reward hacking as a max-min optimization problem over proxy rewards constrained by correlation with the true rewards. 2) We develop a practical algorithm for the max-min problem, which is further extended to linear rewards with improved robustness and interpretability. 3)

Experiment results demonstrate improved robustness and worst-case rewards across four real-world inspired reward hacking environments.

## 2 Preliminaries

**Reinforcement Learning.** A reinforcement learning (RL) problem can be formulated as an infinite-horizon Markov Decision Process (MDP) defined by the tuple $(\mathcal{S}, \mathcal{A}, p, \mu_0, R, \gamma)$, where $\mathcal{S}$ and $\mathcal{A}$ denote the state and action spaces, $p(s' \mid s, a)$ is the transition probability from state $s$ to $s'$ given action $a$, and $\mu_0$ is the initial state distribution. The agent interacts with the environment over discrete time steps $t = 0, 1, 2, \ldots$. At each time step, it selects an action $a_t \in \mathcal{A}$ based on the current state $s_t \in \mathcal{S}$ according to a policy $\pi(a \mid s)$, which defines a distribution over actions conditioned on the state. Upon taking action $a_t$, the agent receives a reward $R(s_t, a_t) \in \mathbb{R}$ and transitions to the next state $s_{t+1}$ according to $p(s_{t+1} \mid s_t, a_t)$. The goal of the agent is to maximize the expected cumulative discounted return:

$$J(\pi, R) = (1 - \gamma) \, \mathbb{E}_\pi \left[ \sum_{t=0}^{\infty} \gamma^t R(s_t, a_t) \right], \tag{1}$$

where $\gamma \in [0, 1)$ is the discount factor, and the expectation is taken over trajectories generated by following policy $\pi$. We define the *state-action occupancy measure* $\mu_\pi$ of a policy $\pi$ as: $\mu_\pi(s, a) = (1 - \gamma) \, \mathbb{E}_\pi \left[ \sum_{t=0}^{\infty} \gamma^t \mathbb{I}\{s_t = s, a_t = a\} \right]$, which represents the discounted visitation frequency of each state-action pair under policy $\pi$. Using the occupancy measure, the return can be equivalently expressed as: $J(\pi, R) = \mathbb{E}_{(s,a) \sim \mu_\pi}[R(s, a)]$.

**Correlated Proxies and Reward Hacking.** Below we give an overview of the recently proposed $r$-correlated proxy framework proposed in [25] for detecting and mitigating reward hacking, which our work is built upon. A detailed discussion of related work on reward hacking and robust reinforcement learning is given in Appendix C. In particular, [25] considers a setting where the agent is given a reference policy $\pi_{\text{ref}}$ and a proxy reward $R_{\text{proxy}}$, while the true reward is hidden. They further assume that the proxy reward is *$r$-correlated* with the true reward under the reference policy, that is:

$$\mathbb{E}_{\mu_{\pi_{\text{ref}}}} \left[ \left( \frac{R_{\text{proxy}} - J(\pi_{\text{ref}}, R_{\text{proxy}})}{\sigma_{R_{\text{proxy}}}} \right) \left( \frac{R_{\text{true}} - J(\pi_{\text{ref}}, R_{\text{true}})}{\sigma_{R_{\text{true}}}} \right) \right] = r, \tag{2}$$

where $\sigma_{R_{\text{proxy}}}^2 = \mathbb{E}_{\mu_{\pi_{\text{ref}}}} \left[ (R_{\text{proxy}} - J(\pi_{\text{ref}}, R_{\text{proxy}}))^2 \right]$ and $\sigma_{R_{\text{true}}}^2 = \mathbb{E}_{\mu_{\pi_{\text{ref}}}} \left[ (R_{\text{true}} - J(\pi_{\text{ref}}, R_{\text{true}}))^2 \right]$ are the variances of the proxy and true rewards, respectively, under the reference policy. Reward hacking is said to occur when a policy $\pi$ optimized for an $r$-correlated proxy reward achieves lower true reward than the reference policy: $J(\pi, R_{\text{true}}) < J(\pi_{\text{ref}}, R_{\text{true}})$. To mitigate reward hacking, [25] proposes Occupancy-Regularized Policy Optimization (ORPO) to optimize a regularized policy objective given below, which is shown to provide a lower bound on improvement in true reward:

$$\max_\pi \; J(\pi, R_{\text{proxy}}) - \lambda \sqrt{\chi^2(\mu_\pi \, \| \, \mu_{\pi_{\text{ref}}})}, \tag{3}$$

where $\chi^2(\mu_\pi \, \| \, \mu_{\pi_{\text{ref}}})$ denotes the $\chi^2$-squared divergence between the occupancy measures of $\pi$ and $\pi_{\text{ref}}$, and the regularization strength $\lambda$ is set as: $\lambda = \sigma_{R_{\text{proxy}}} \sqrt{1 - r^2}$. This encourages the learned policy to stay close to the reference distribution when the proxy reward is weakly correlated with the true reward.

## 3 Method

In this section, we discuss our robust policy optimization approach for mitigating reward hacking. In contrast to regularization-based methods such as ORPO, we consider a max-min formulation that identifies a robust policy with respect to the worst-case reward across all reward functions that are $r$-correlated with the proxy reward. We further extend our framework to settings where the reward function is a linear combination of known features with unknown weights. Our approach effectively leverages this structural information, when known a priori, to improve both robustness and interpretability, a task that is particularly challenging for regularization-based techniques.

## 3.1 Max-Min Policy Optimization

Similar to ORPO, we assume that the agent is given a proxy reward $R_{\text{proxy}}$ and a reference policy $\pi_{\text{ref}}$, while the true reward is hidden. Rather than regularizing the policy under a fixed proxy reward, we consider the *entire space of rewards* $\mathcal{R}_{\text{corr}}$ that satisfy the correlation constraint with respect to a known proxy reward, as defined in Equation 4:

$$\mathcal{R}_{\text{corr}} = \left\{ R : (s,a) \to \mathbb{R} \; \middle| \; \mathbb{E}_{\mu_{\pi_{\text{ref}}}} \left[ \frac{R-M}{V} \cdot R_{\text{proxy}} \right] = r, \; J(\pi_{\text{ref}}, R) = M, \; \sigma_R^2 = V^2 \right\}. \quad (4)$$

$M$ and $V$ denote the fixed mean and standard deviation of the reward function $R$ under the reference policy $\pi_{\text{ref}}$. For simplicity, we define $R_{\text{proxy}}$ to be the normalized proxy reward $R_{\text{proxy}}(s,a) := \frac{\tilde{R}_{\text{proxy}}(s,a) - J(\pi_{\text{ref}}, \tilde{R}_{\text{proxy}})}{\sigma_{\tilde{R}_{\text{proxy}}}}$, where $\tilde{R}_{\text{proxy}}$ is the original (unnormalized) proxy reward. After normalization, we have $J(\pi_{\text{ref}}, R_{\text{proxy}}) = 0$ and $\text{Var}_{\mu_{\pi_{\text{ref}}}}(R_{\text{proxy}}) = 1$, which simplifies the correlation constraint in Equation 4. The hyperparameter $r$ controls the degree of alignment between the proxy and true reward. It allows us to interpolate between strong robustness (small $r$) and high proxy fidelity (large $r$), enabling a principled robustness-accuracy trade-off. We remark that it is without loss of generality to consider fixed $M$ and $V$, which we will further elaborate on later.

We propose a *worst-case optimization framework* where the policy is trained to maximize expected performance under the least favorable reward within $\mathcal{R}_{\text{corr}}$. Assuming that the true reward lies somewhere within this set, this approach improves robustness by ensuring that the policy does not overfit to any single optimistic interpretation of the proxy reward. Formally, the objective becomes a max-min problem:

$$\max_{\pi} \min_{R \in \mathcal{R}_{\text{corr}}} J(\pi, R) = \max_{\pi} \min_{R \in \mathcal{R}_{\text{corr}}} \mathbb{E}_{(s,a) \sim \mu_\pi}[R(s,a)]. \quad (5)$$

However, a challenge arises: the objective $\mathbb{E}_{\mu_\pi}[R(s,a)]$ depends on the state-action occupancy $\mu_\pi$, whereas the constraints defining $\mathcal{R}_{\text{corr}}$ are expressed in terms of $\mu_{\pi_{\text{ref}}}$. This mismatch complicates direct optimization. To resolve this, we apply a *change-of-measure* technique [28, 29] to rewrite the expectation under $\mu_{\pi_{\text{ref}}}$. Specifically, let $L(s,a)$ denote the Radon-Nikodym derivative: $L(s,a) = \frac{\mu_\pi(s,a)}{\mu_{\pi_{\text{ref}}}(s,a)}$. By definition, $L(s,a) \geq 0$ and $\mathbb{E}_{\mu_{\pi_{\text{ref}}}}[L(s,a)] = 1$. Applying the change-of-measure formula, we can express the return as: $\mathbb{E}_{\mu_\pi}[R(s,a)] = \int_{\mathcal{S} \times \mathcal{A}} \mu_\pi(s,a) R(s,a)\, d(s,a) = \int_{\mathcal{S} \times \mathcal{A}} \mu_{\pi_{\text{ref}}}(s,a) \frac{\mu_\pi(s,a)}{\mu_{\pi_{\text{ref}}}(s,a)} R(s,a)\, d(s,a) = \mathbb{E}_{\mu_{\pi_{\text{ref}}}}[L(s,a) R(s,a)]$. Thus, both the objective and the constraints can be rewritten as expectations with respect to the reference distribution $\mu_{\pi_{\text{ref}}}$.

For notational simplicity, we will suppress the variables $(s,a)$ and write for example, $L$ to denote $L(s,a)$ and $R$ to denote $R(s,a)$. Under this reparameterization, the inner minimization in Equation 5 can be reformulated as:

$$\min_{R \in \mathcal{R}_{\text{corr}}} \mathbb{E}_{\mu_{\pi_{\text{ref}}}}[L \cdot R]. \quad (6)$$

Although the feasible set in Problem 6 is not convex due to the equality constraint on the variance, we still derive an optimal solution using a Lagrangian formulation. Our approach leverages tools from duality theory, commonly used in robust optimization [30, 31]. We further justify the validity of our solution in Appendix D.2. Specifically, the Lagrangian functional associated with this problem is defined as: $l_0(\lambda_1, \lambda_2, \lambda_3, R) = \mathbb{E}_{\mu_{\pi_{\text{ref}}}}[L \cdot R - \lambda_1 \frac{R-M}{V} \cdot R_{\text{proxy}} - \lambda_2 R - \lambda_3 R^2] + \lambda_1 r + \lambda_2 M + \lambda_3 (M^2 + V^2)$, where $\lambda_1, \lambda_2, \lambda_3$ are the Lagrange multipliers corresponding to the correlation constraint, mean constraint, and variance constraint, respectively. Then the original problem in Equation 6 is equivalent to the following problem:

$$\max_{\lambda_1, \lambda_2, \lambda_3} \min_{R \in \mathcal{R}_{\text{corr}}} l_0(\lambda_1, \lambda_2, \lambda_3, R). \quad (7)$$

We now solve the inner minimization problem in Equation 7 by finding the optimal $R$ for fixed dual variables $(\lambda_1, \lambda_2, \lambda_3)$. Taking the functional derivative of the Lagrangian $l_0$ with respect to $R(s,a)$ gives: $\frac{\partial l_0}{\partial R} = \mu_{\pi_{\text{ref}}}(s,a)[(L - \lambda_1 \frac{R_{\text{proxy}}}{V} - \lambda_2) - 2\lambda_3 R]$. When $\mu_{\pi_{\text{ref}}}(s,a) > 0$, setting the derivative of the Lagrangian to zero yields the optimal adversarial reward function:

$$R^*(s,a) = \frac{L(s,a) - \lambda_1 \frac{R_{\text{proxy}}}{V} - \lambda_2}{2\lambda_3}. \quad (8)$$

However, for state-action pairs where $\mu_{\pi_{\mathrm{ref}}}(s, a) = 0$, i.e., those not visited under the reference policy, the correlation and moment constraints become vacuous. In these regions, the adversarial reward $R^*(s, a)$ can be driven arbitrarily poor, reflecting that no constraint prevents the adversary from assigning highly penalizing values to rarely visited or unobserved state-action pairs. Nevertheless, consider the case where $\mu_{\pi_{\mathrm{ref}}}(s, a) > 0$, we can substitute the optimal $R^*$ from Equation 8 into the Lagrangian $l_0$ and get the dual objective. After some process detailed in Appendix D.1, we get the optimal solution to the inner problem (6), so the original max-min problem (5) reduces to:

$$\max_{\pi} \ r \cdot V \cdot \mathbb{E}_{\mu_{\pi}}[R_{\mathrm{proxy}}] - V \cdot \sqrt{1 - r^2}\sqrt{\chi^2(\mu_{\pi} \parallel \mu_{\pi_{\mathrm{ref}}}) - \mathbb{E}^2_{\mu_{\pi}}[R_{\mathrm{proxy}}]} + M. \tag{9}$$

Thus, the final policy optimization objective becomes maximizing the proxy reward, regularized by a penalty that depends on the distributional shift between $\mu_{\pi}$ and $\mu_{\pi_{\mathrm{ref}}}$ and the expectation of the current policy under proxy reward $\mathbb{E}_{\mu_{\pi}}[R_{\mathrm{proxy}}]$, and the correlation strength $r$. We observe that the constants $M$ and $V$ do not affect the optimal policy: while they influence the absolute value of the worst-case reward for a given policy $\pi$, they only apply a linear transformation (scaling by $V$ and shifting by $M$) and do not change the relative ordering of policies. Therefore, for simplicity, we set $V = 1$ and $M = 0$ in our implementation. This also provides a fair way to compare the worst-case rewards of different policies. Notice that the optimization objective in Equation 9 closely resembles the ORPO objective proposed in Equation 3. However, there are two key differences: (1) our regularization strength is $\frac{\sqrt{1-r^2}}{r}$ instead of $\sigma_{R_{\mathrm{proxy}}}\sqrt{1 - r^2}$, and (2) our penalty term is $\chi^2(\mu_{\pi} \parallel \mu_{\pi_{\mathrm{ref}}}) - \mathbb{E}^2_{\mu_{\pi}}[R_{\mathrm{proxy}}]$ rather than simply $\chi^2(\mu_{\pi} \parallel \mu_{\pi_{\mathrm{ref}}})$. The proof that $\chi^2(\mu_{\pi} \parallel \mu_{\pi_{\mathrm{ref}}}) - \mathbb{E}^2_{\mu_{\pi}}[R_{\mathrm{proxy}}] \geq 0$ holds can be found in Appendix D.3. A detailed comparison between our policy gradient and that of ORPO is provided in Appendix D.8.

## 3.2 Structured Reward Spaces via Feature Linearization

A natural concern with worst-case optimization is *over-conservatism*: if the reward uncertainty set $\mathcal{R}_{\mathrm{corr}}$ is too broad, the resulting policy may become overly cautious or deviate from realistic task objectives. Additionally, the learned worst-case rewards may themselves be implausible or uninterpretable. To address these issues, we introduce *structure* into the reward space by assuming that all rewards are *linear combinations of known features*. Specifically, we assume: $R(s, a) = \boldsymbol{\theta}^\top \boldsymbol{\phi}(s, a)$, where $\boldsymbol{\phi}(s, a) = [\phi_1(s, a), \phi_2(s, a), \dots, \phi_M(s, a)]^\top \in \mathbb{R}^M$ denotes a vector of $M$ known or engineered feature functions, and $\boldsymbol{\theta} = [\theta_1, \theta_2, \dots, \theta_M]^\top \in \mathbb{R}^M$ represents the uncertain feature weights. The linearization yields two key benefits: 1) **Realism and Interpretability:** In many real-world tasks, reward functions are naturally approximated as linear combinations over interpretable features. For example, in a traffic control environment, features might include total commute time, vehicle speed, acceleration, and inter-vehicle headway distances. 2) **Better-Constrained Robustness:** By restricting uncertainty to structured, feature-based rewards, the worst-case optimization problem becomes more grounded and avoids pathological, unrealistic reward functions.

In this section, we assume that the agent is aware of the set of features but not their true weights. We show that our robust optimization framework can be naturally extended to incorporate the structure in rewards to improve robustness. In our experiments, we further demonstrate that linear rewards help interpret a policy's performance even when it is trained without such prior knowledge. Under our assumption, the uncertainty set reduces to the set of feature weights $\boldsymbol{\theta} \in \mathbb{R}^M$ satisfying:

$$\mathcal{R}_{\mathrm{corr}}^{\mathrm{lin}} = \left\{ \boldsymbol{\theta} \in \mathbb{R}^M \ \middle| \ \mathbb{E}_{\mu_{\pi_{\mathrm{ref}}}}[\boldsymbol{\theta}^\top \boldsymbol{\phi} \cdot R_{\mathrm{proxy}}] = r, \ \mathbb{E}_{\mu_{\pi_{\mathrm{ref}}}}[\boldsymbol{\theta}^\top \boldsymbol{\phi}] = 0, \ \mathbb{E}_{\mu_{\pi_{\mathrm{ref}}}}[(\boldsymbol{\theta}^\top \boldsymbol{\phi})^2] = 1 \right\}. \tag{10}$$

To simplify the analysis, we assume without loss of generality that the worst-case reward $R(s, a) = \boldsymbol{\theta}^\top \boldsymbol{\phi}(s, a)$ is normalized to have zero mean and unit variance under the reference policy $\pi_{\mathrm{ref}}$. This corresponds to setting $M = 0$ and $V = 1$, which, as shown in our earlier derivation, does not affect the resulting optimal policy. As before, $R_{\mathrm{proxy}}$ denotes the normalized proxy reward under $\pi_{\mathrm{ref}}$, satisfying $\mathbb{E}_{\mu_{\pi_{\mathrm{ref}}}}[R_{\mathrm{proxy}}] = 0$ and $\mathrm{Var}_{\mu_{\pi_{\mathrm{ref}}}}[R_{\mathrm{proxy}}] = 1$.

We now derive the corresponding max-min optimization under the structured reward assumption:

$$\max_{\pi} \min_{\boldsymbol{\theta} \in \mathcal{R}_{\mathrm{corr}}^{\mathrm{lin}}, \boldsymbol{\theta} \geq 0} \mathbb{E}_{(s,a) \sim \mu_{\pi}} \left[ \boldsymbol{\theta}^\top \boldsymbol{\phi}(s, a) \right]. \tag{11}$$

Similar to previous steps, we introduce the Radon-Nikodym derivative $L(s, a) = \frac{\mu_\pi(s,a)}{\mu_{\pi_{\text{ref}}}(s,a)}$, use a change-of-measure, and define the Lagrangian functional for the inner minimization in Equation 11 as: $l_1(\lambda_1, \lambda_2, \lambda_3, \boldsymbol{\theta}) = \boldsymbol{\theta}^\top \left( \sum_{(s,a)} u_{\lambda_1, \lambda_2}(s, a)\boldsymbol{\phi}(s, a) \right) - \lambda_3 \boldsymbol{\theta}^\top Q \boldsymbol{\theta} + \lambda_1 r + \lambda_3$, where $u_{\lambda_1, \lambda_2} = \mu_\pi - \lambda_1 \mu_{\pi_{\text{ref}}} R_{\text{proxy}} - \lambda_2 \mu_{\pi_{\text{ref}}}$, $Q = \sum_{(s,a)} \mu_{\pi_{\text{ref}}}(s, a)\boldsymbol{\phi}(s, a)\boldsymbol{\phi}(s, a)^\top$. A detailed derivation can be found in Appendix D.4. Note that $Q$ is positive semi-definite since it is a sum of outer products $\boldsymbol{\phi}(s, a)\boldsymbol{\phi}(s, a)^\top$ weighted by non-negative coefficients (occupancy measure of $\pi_{\text{ref}} \geq 0$). Then solving the inner minimization over $\boldsymbol{\theta}$ in Equation 11 is equivalent to solving:

$$\max_{\lambda_1, \lambda_2, \lambda_3} \min_{\boldsymbol{\theta} \geq 0} \quad l_1(\lambda_1, \lambda_2, \lambda_3, \boldsymbol{\theta}) = \boldsymbol{\theta}^\top \left( \sum u_{\lambda_1, \lambda_2} \boldsymbol{\phi} \right) - \lambda_3 \boldsymbol{\theta}^\top Q \boldsymbol{\theta} + \lambda_1 r + \lambda_3. \tag{12}$$

Notice that $l_1(\lambda_1, \lambda_2, \lambda_3, \boldsymbol{\theta})$ is a convex quadratic function of $\boldsymbol{\theta}$ (assuming $\lambda_3 \leq 0$) subject to linear inequality constraints $\boldsymbol{\theta} \geq 0$. Thus, the original problem is a standard convex quadratic program (QP) with non-negativity constraints [32]. However, it is not possible to derive a universal closed-form solution for the optimal $\boldsymbol{\theta}^*$ under arbitrary $Q$. To further simplify the problem and obtain a closed-form solution, we transform the feature vector $\boldsymbol{\phi}$ into a whitened version $\tilde{\boldsymbol{\phi}}$ such that the matrix $Q$ becomes the identity matrix $I$ and we formally show this in Appendix D.5. Specifically, we perform a whitening transformation using the Cholesky decomposition [32]. Let $W = Q^{-\frac{1}{2}}, \tilde{\boldsymbol{\phi}}(s, a) = W\boldsymbol{\phi}(s, a)$, where $Q^{-\frac{1}{2}}$ denotes a matrix square root of $Q^{-1}$ (which exists since $Q$ is positive semi-definite and non-singular assuming $\exists (s, a)$ such that $\mu_{\pi_{\text{ref}}}(s, a) > 0$). Then the original problem in Equation 12 can be further simplified into:

$$\max_{\lambda_1, \lambda_2, \lambda_3} \min_{\tilde{\boldsymbol{\theta}} \geq 0} \quad l_1(\lambda_1, \lambda_2, \lambda_3, \tilde{\boldsymbol{\theta}}) = \tilde{\boldsymbol{\theta}}^\top \left( \sum_{(s,a)} u_{\lambda_1, \lambda_2}(s, a)\tilde{\boldsymbol{\phi}}(s, a) \right) - \lambda_3 \tilde{\boldsymbol{\theta}}^\top \tilde{\boldsymbol{\theta}} + \lambda_1 r + \lambda_3. \tag{13}$$

where we now optimize over the parameter $\tilde{\boldsymbol{\theta}}$ using the transformed features $\tilde{\boldsymbol{\phi}}$. For notational simplicity, we will drop the tilde and henceforth use $\boldsymbol{\phi}$ to represent the whitened feature $\tilde{\boldsymbol{\phi}}$, and $\boldsymbol{\theta}$ to represent the whitened weights $\tilde{\boldsymbol{\theta}}$. Then we can get a closed-form solution (we detail the steps in Appendix D.6) for optimal $\boldsymbol{\theta}^*$ as: $\boldsymbol{\theta}^* = \max \left( 0, -\frac{\sum_{(s,a)} u_{\lambda_1, \lambda_2}(s,a)\boldsymbol{\phi}(s,a)}{2\lambda_3} \right)$, where the $\max(\cdot, 0)$ is applied elementwise. Details for solving the outer maximization in Equation 13 can be found in Appendix D.7. After obtaining the optimal dual variables $(\lambda_1^*, \lambda_2^*, \lambda_3^*)$, we can substitute them back into Equation 11 and solve the outer maximization over the policy $\pi$ using standard reinforcement learning algorithms, such as PPO [33].

**ORPO with Linear Awards.** While ORPO provides a general guarantee based on occupancy measure regularization, it does not exploit any structural assumptions about the reward function. In particular, even when the true reward is linear in a set of features, ORPO does not explicitly incorporate this structure into its policy optimization or theoretical analysis. While the lower bound (Theorem 5.1 in [25]) continues to hold, it is unclear how to leverage this structure to obtain a tighter lower bound or to guide policy updates more effectively. This suggests a missed opportunity: by explicitly modeling the reward as a linear function, it becomes possible to derive stronger guarantees, interpret worst-case reward directions, and efficiently optimize against them. Our Linear Maxmin method fills this gap by parameterizing reward uncertainty directly in the space of reward weights, enabling both robustness and greater transparency.

### 3.3 Implementation Details and Algorithms

A core step in both our algorithms and ORPO is to estimate the Radon-Nikodym derivative $L(s, a)$. To this end, we follow prior works [25, 34, 35] and train a discriminator network. Specifically, we use a discriminator architecture identical to that in [25], denoted by $d_\phi(s, a)$, which is optimized according to:

$$\phi = \arg\min_\phi \; \mathbb{E}_{\mu_{\pi_{\text{ref}}}}[\log(1 + e^{d_\phi(s,a)})] + \mathbb{E}_{\mu_\pi}[\log(1 + e^{-d_\phi(s,a)})]. \tag{14}$$

It is known that the optimal discriminator satisfies $d^*(s, a) = \log \frac{\mu_\pi(s,a)}{\mu_{\pi_{\text{ref}}}(s,a)}$ and we estimate $L(s, a)$ as $\tilde{L}(s, a) = \exp d_\phi(s, a)$ with $d_\phi(s, a) \approx d^*(s, a)$. As discussed in Section 3.1, if the policy $\pi$ visits state-action pairs that the reference policy $\pi_{\text{ref}}$ rarely or never visits, the adversarial reward can be arbitrarily poor. In theory, the estimated $\tilde{L}(s, a)$ is expected to grow arbitrarily large in this

case, which should discourage the learned policy from exploiting such regions. However, we observe empirically (Section 4.2) that the ORPO policy still visits some of these low-coverage regions under $\pi_{\text{ref}}$. This is because in the original ORPO implementation[1], the discriminator is not fully optimized during policy learning. Specifically, the discriminator receives only a small number of gradient updates per reinforcement learning iteration, resulting in underfitting and inaccurate estimates of the Radon-Nikodym derivative $\tilde{L}(s, a)$. To address this, we substantially increase the number of gradient updates per iteration and carefully tune the learning rate. Our goal is to strike a practical balance between training time and discriminator quality, which we further discuss in Appendix E.1.

To compute the final objective for our Max-Min policy in Equation 9, we estimate the $\chi^2$ divergence, the normalized proxy reward $R_{\text{proxy}}$, and the first and second moments $\mathbb{E}_{\mu_\pi}[R_{\text{proxy}}]$ and $\mathbb{E}^2_{\mu_\pi}[R_{\text{proxy}}]$. These components together define the robust optimization objective used to update the policy. A simplified Max-Min policy optimization procedure is outlined in Algorithm 1. We provide detailed descriptions of each estimation step, as well as the complete algorithmic implementation in Appendix E.2. Corresponding derivations and implementation details for the Linear Max-Min variant are included in Appendix E.3.

---

**Algorithm 1** Max-Min Policy Optimization (Simplified)

---

1: Initialize policy parameters $\theta$
2: Initialize reference policy $\pi_{\text{ref}}$ and collect trajectories
3: Estimate mean and variance of the proxy reward under $\pi_{\text{ref}}$
4: **for** each iteration **do**
5:     Collect trajectories from current policy $\pi_\theta$
6:     Normalize the proxy rewards for state-action pairs in the collected trajectories
7:     Estimate the expected proxy reward and its second moment under the current policy
8:     Estimate the discriminator using Equation (14) and $\chi^2$ divergence between $\mu_\pi$ and $\mu_{\pi_{\text{ref}}}$
9:     Update the policy using PPO to maximize the Max-Min objective in Equation (9)
10: **end for**

---

## 4 Experiment

### 4.1 Experiment Setup

We evaluate our method across four realistic benchmark environments: *Traffic*, *Pandemic*, *Glucose Monitoring*, and *Tomato Watering GridWorld*. These environments were originally proposed in [36, 5] and represent diverse forms of proxy reward hacking, including misweighting, ontological mismatch, and scope misalignment [36]. Each setting presents unique challenges in reward specification and policy robustness. A detailed description of the environments and their respective reward structures is provided in Appendix E.4. In each of the four environments, we train policies using both our `Max-Min` and `Linear Max-Min` optimization algorithms. For baselines, we compare against the `ORPO` policy. To isolate the impact of discriminator training, we also include an ablation: `ORPO*`, where we train the ORPO policy using the same full discriminator training schedule as in our algorithms. This variant shares the same architecture and optimization settings as the original ORPO, differing only in the extent of discriminator training. Including this baseline allows us to evaluate the specific contribution of discriminator optimization to policy robustness. We include more detailed experimental settings in Appendix E.5 and a discussion of training time and complexity of all algorithms in Appendix E.6.

As for evaluation metrics, we report both the expected proxy and true rewards, along with the expected worst-case reward as described in Section 3.1. Note that some policies may visit state-action pairs that are not covered by the reference policy $\pi_{\text{ref}}$. In such cases, we exclude those trajectories and report the occupancy measure of the unseen state-action pairs. Additionally, we evaluate each policy using two variants of the expected linear worst-case reward introduced in Section 3.2. The first uses only the features present in the proxy reward, while the second variant, denoted *Linear Worst\**, leverages features from the true reward, some of which remain unseen during training. This setup mimics a more realistic real-world scenario in which the true reward function may depend on features not explicitly modeled at training time. Comparing performance under this setting allows us to assess the robustness of each policy to unseen or misaligned reward structures. All rewards are normalized with respect to the reference policy $\pi_{\text{ref}}$ to ensure a consistent scale across metrics, enabling fair and

---

[1]https://github.com/cassidylaidlaw/orpo/tree/main

Table 1: Evaluation results on Traffic and Pandemic environments. All policies are trained using **only the proxy reward**. In Traffic, the proxy reward is based on *vel*, *accel*, *headway* (1, 1, 0.1), while the true reward uses *commute*, *accel*, *headway* (1, 1, 0.1). In Pandemic, the proxy reward includes *infection*, *lower stage*, *smooth changes* (10, 0.1, 0.01), while the true reward additionally includes *political* with weight 10 after *infection*. We report $\theta$ in the same order as feature weights. **Occ** denotes total occupancy over state-action pairs unseen by $\pi_{\text{ref}}$, where discriminator outputs infinity.

| Env | Traffic | | | | | |
|---|---|---|---|---|---|---|
| Method | True | Proxy | Worst | Linear Worst* ($\theta$) | Linear Worst* ($\theta$) | Occ ↓ |
| ORPO | 16.93 | 3.31 | -1.97e+04 | -0.68 (0.71, 0.21, 0.69) | -0.81 (0.63, 0.12, 0.97) | 3.71e-04 |
| ORPO* | 10.31 | 1.32 | -1.33e+04 | -0.42 (0.46, 0.18, 0.86) | -0.44 (0.58, 0.06, 0.81) | 1.90e-05 |
| Max-Min | 12.64 | 3.64 | -270.84 | -0.07 (0.01, 0.02, 0.96) | -0.07 (0.001, 0.02, 0.99) | 0 |
| Linear Max-Min | 16.36 | 2.53 | -1.19e+04 | 0.21 (0.64, 0.07, 0.76) | -0.12 (0.91, 0.01, 0.67) | 0 |
| Env | Pandemic | | | | | |
| Method | True | Proxy | Worst | Linear Worst ($\theta$) | Linear Worst* ($\theta$) | |
| ORPO | -0.91 | 1.81 | -5.30e+06 | -2.42 (0.23, 0.95, 0.17) | -2.63 (0.02, 0.95, 0.92, 0.08) | |
| ORPO* | 1.24 | 1.24 | -4.42e+06 | -1.35 (0.25, 0.97, 0.13) | -1.35 (0.25, 0, 0.97, 0.13) | |
| Max-Min | 1.15 | 1.15 | -65.69 | -1.11 (0.14, 0.99, 0.01) | -1.11 (0.14, 0, 0.99, 0.01) | |
| Linear Max-Min | 2.61 | 7.56 | -6.83e+05 | 0.66 (0.001, 0.23, 0.02) | -0.17 (0.01, 0.97, 0.22, 0.09) | |

meaningful comparisons. Note that all worst-case rewards are reported using the fixed correlation level $r$ specified during training: $r = 0.3$ for Traffic, $r = 0.7$ for Pandemic, with values for other environments provided in Appendix E.5.

## 4.2 Results

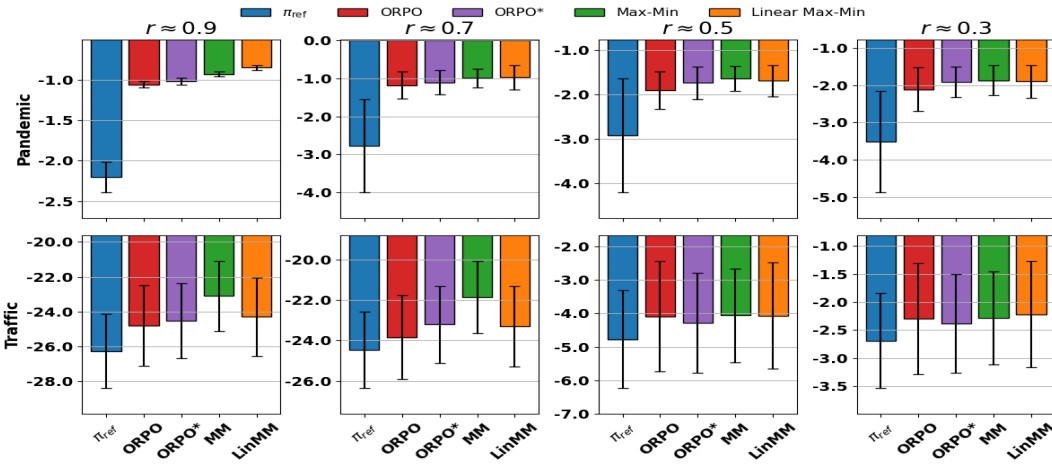

Figure 1: Mean reward and standard deviation under sampled $\theta$ and true reward features at different proxy–true reward correlation levels $r$ for the Traffic and Pandemic environments. Our methods (`Maxmin` and `Linear Maxmin`) yield more stable and higher average performance across all choices of $r$.

**Worst-Case Performance.** Table 1 presents the evaluation results on the Traffic and Pandemic environments. Additional results for other environments are provided in Appendix F. Our `Max-Min` and `Linear Max-Min` policies achieve better expected worst-case performance under both general and linear adversarial rewards, while remaining competitive with baselines in terms of expected true and proxy rewards. Notably, the `Max-Min` policy attains the highest expected worst-case return, followed by `Linear Max-Min`. Conversely, `Linear Max-Min` yields the highest expected linear worst-case reward, followed by `Max-Min`, demonstrating the robustness of both approaches under worst-case scenarios. For the Linear Worst* evaluation, which uses reward features unseen during training, we observe minimal degradation in `Max-Min` policy's performance, indicating its strong robustness to feature variation. In contrast, the performance of `Linear Max-Min` declines in this case, suggesting its advantage diminishes when prior assumptions about feature structure are inaccurate.

We find that `ORPO*` exhibits better worst-case performance than the original `ORPO`. In particular, training the discriminator more thoroughly significantly reduces the occupancy of state-action pairs that are not visited by the reference policy, indicating that more accurate estimation of the Radon–Nikodym derivative leads to improved policy robustness. Notably, in the Pandemic environment, we observe no such unvisited state-action pairs, and the discriminator outputs remain small across all policies. This

could be due to either the discriminator network not being fully optimized or its inability to capture rare events that fall outside the support of $\pi_{\text{ref}}$. Developing more reliable techniques for handling such rare or unseen state-action pairs remains an open direction for future work.

We also report the adversarial weight vectors $\boldsymbol{\theta}$ for each policy. These weights reveal which features are most vulnerable to proxy exploitation under the learned policy and can be used to diagnose and revise the proxy reward function, thereby improving robustness. This highlights the interpretability benefits of our framework. Moreover, several patterns emerge from the results. In the Traffic environment, first, we observe a clear dominance of the headway feature, with all methods assigning it the highest weight. This suggests that headway is the most critical component exposed to reward hacking under correlation constraints. Second, the acceleration feature is consistently downweighted across all methods. This indicates that acceleration may be less prone to exploitation or already well aligned with the reference policy. Third, the velocity feature is moderately emphasized by `Linear Max-Min` and `ORPO` (e.g., $0.64$ and $0.71$), while `Max-Min` nearly suppresses it ($0.01$). This contrast suggests that `Linear Max-Min` anticipates some vulnerability from velocity deviations, while `Max-Min` focuses almost entirely on headway. In the Pandemic environment, first, both `ORPO*` and `Max-Min` assign zero weight to the political feature. This occurs because the expected feature value under their policies is exactly zero, making the correlation constraint inactive for that dimension. Interestingly, this feature plays a significant role in the adversarial rewards for both `ORPO` and `Linear Max-Min`, with their corresponding $\boldsymbol{\theta}$ assigning non-negligible weight to it (e.g., $0.95$ and $0.97$ respectively). This suggests that these policies expose themselves to vulnerability in feature dimensions that are entirely ignored by `Max-Min` and `ORPO*`. Second, the lower stage feature consistently receives the highest weight across all methods, indicating it is the most sensitive component under proxy misalignment.

**Robustness Across Correlation Levels.** To further assess the robustness of each policy across a broader range of proxy–true correlation scenarios, we also compute the Linear Worst* for each policy under varying $r$ values. Specifically, for each $r$, we sample 1000 vectors $\boldsymbol{\theta}$ such that $\boldsymbol{\theta} \in \mathcal{R}_{\text{corr}}^{\text{lin}}$, and report the average return and variance achieved by each policy over these sampled rewards. Importantly, the variation in $r$ is applied **only during evaluation**; all policies are fixed and trained using the specific $r$ values reported in Appendix E.5. Unlike evaluations that only consider several reward functions, this approach evaluates policy performance across the entire reward set $\mathcal{R}_{\text{corr}}^{\text{lin}}$, providing a more comprehensive measure of robustness and better reflecting real-world scenarios where the true reward and correlation $r$ are unknown.

Figure 1 shows the average reward and variance achieved by each method under different levels of proxy–true reward correlation $r$. As expected, the base policy $\pi_{\text{base}}$ (blue) performs the worst across all correlation levels in both environments. In Traffic, its variance is relatively small, suggesting consistently poor but stable behavior. In contrast, variance is highest in the Pandemic environment, indicating increased policy fragility. Notably, `ORPO*` (purple) consistently achieves lower variance than `ORPO` (red) across both environments and outperforms it in terms of average reward at $r \approx 0.9$ and $r \approx 0.7$ in Traffic, and across nearly all $r$ values in Pandemic. This underscores the importance of accurate discriminator training for improving both stability and robustness. `Max-Min` (green) demonstrates the highest average reward and lowest variance across a wide range of $r$ values in both environments, showing strong resilience to reward misspecification. While `Linear Max-Min` (orange) achieves the best performance at specific correlation levels, particularly $r \approx 0.3$ in Traffic and $r \approx 0.7$–$0.9$ in Pandemic. As $r$ decreases and the proxy becomes less informative, differences in average reward among methods shrink, while variance increases. These results highlight the significance of variance control in low-correlation regimes and demonstrate that `Max-Min` and `Linear Max-Min` offer robust and stable performance under high uncertainty.

## 5 Conclusion

In this work, we propose a robust policy optimization framework that explicitly accounts for reward hacking by training policies against the worst-case proxy reward drawn from a correlation-constrained uncertainty set. Our approach formalizes reward hacking as a robust optimization problem and introduces both a Max-Min formulation with a closed-form adversarial reward and a Linear Max-Min variant that further improves interpretability and tractability. We develop efficient algorithms and empirically validate our methods across diverse environments known to exhibit reward hacking behavior. Our results demonstrate that both Max-Min and Linear Max-Min policies achieve stronger worst-case performance and improved stability compared to prior baselines such as ORPO.

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
