# OpenReview forum: "Robust Optimization for Mitigating Reward Hacking with Correlated Proxies"
_NeurIPS.cc/2025/Conference — Submitted to NeurIPS 2025_

### Official Review · Reviewer_EXFr · 2025-06-17

**Clarity:** 3
**Significance:** 4
**Originality:** 3
**Rating:** 5
**Confidence:** 4

**Summary:**

This paper constructs a setup where a reinforcement learner is informed: the correlation between your real reward and the reward you see equals X if you're following this reference policy Pi; given that information, act to optimize the worst-case version of the real reward. The paper develops an efficient algorithm for how to train such a reinforcement learner. Their derivation produces an algorithm that is similar to a recent algorithm (ORPO), but with important differences. They also study how performance can improve when the reinforcement learner is also informed: your real reward is some (positive) linear combination of the following features, and this algorithm ends up looking completely different from ORPO. They also develop a critical improvement to a subroutine of ORPO that applies to their algorithm as well. Their empirical results show that their algorithms accomplish what they aim to accomplish--improving the worst-case performance.

**Questions:**

"Although the original Glucose simulator provides multiple candidate features related to patient health, selecting an appropriate feature combination without prior knowledge of clinical intent is nontrivial. In the Tomato environment, the reward structure is similarly difficult to express in a clean feature-based form suitable for linear modeling." What do you conclude from this? That conclusion should make it into the main paper.

On Tomato, why does Max-min have worse "worst case" reward than ORPO* when it is explicitly maximizing that? Why does Max-min enter states that have zero occupancy under the reference policy?

**Ethical Concerns:**

["NO or VERY MINOR ethics concerns only"]

**Final Justification:**

Thank you to the authors and other reviewers for the conversation. I am keeping my score the same.

**Limitations:**

Not quite. I think limitations should be discussed in the main paper, not just the appendix. I think they should include the challenge of constructing good features, and the challenge of ensuring good coverage of the features by the reference policy. The latter is not just a logistical challenge--when some features vary, that means something bad has happened!

**Quality:**

3

**Strengths And Weaknesses:**

The problem that the authors are trying to solve is a critically important one, and they make meaningful theoretical progress with this work. I would characterize the empirical results as a strong sanity check--they don't really prove that this method will improve the performance of frontier RL systems, but in my view, a) the theoretical results are more than enough to carry this paper, and b) the empirical results support the position that the algorithm is practical and potentially useful.

Here are some weaknesses that I believe can be easily fixed:

First, a mathematical mistake.

> "which exists since Q is positive semi-definite and non-singular assuming ∃(s, a) such that μπref (s, a) > 0"

That condition is not enough for non-singularity. That just ensures that Q is PSD with rank at least 1. There need to exist n state-action pairs with positive reference occupancy measure and linearly independent feature vectors. If that condition isn't met, then the data doesn't constrain how the worst-case R changes along a direction that is outside the space spanned by feature vectors of states with positive reference occupancy measure. This is a fairly important error that could easily have practical consequences. So in addition to this requiring a correction, the paper should discuss when we can expect all the features to exhibit variation under the reference policy.

Next, some of the presentation confused me and required me to dig through 3 sections of the appendix. Here was my process:

> "After obtaining the optimal dual variables (λ∗1 , λ∗2 , λ∗3 ), we can substitute them back  into Equation 11 and solve the outer maximization over the policy π using standard reinforcement learning algorithms, such as PPO [33]."

I don't understand how this works. $\lambda_i^\star$ depends on $theta^\star$, which depends on $u$, which depends on $\mu_\pi$, which depends on $\pi$. How can we solve for $\lambda_i^\star$ first and then solve for $\pi$ after? I'm looking up at the original formulation (without feature linearization) to see if I can understand how the corresponding problem is solved there, and I can't work it out from just looking at the main text. From equation 8, $R^\star$ depends on $L$, which depends on $\pi$. Then "we can substitute the optimal R∗ from Equation 8 into the Lagrangian l0 and get the dual objective. After some process detailed in Appendix D.1," we get the formulation max_pi [expression].

Going to the appendix, I see that this process depends on a few identities at line 1054. $(E_{\mu_{\pi_{ref}}} [L] = 1$, and two others). So I'm satisfied that this works for the original formulation. But in Appendix D.6 and D.7, I don't see any similar trick. The gradients for the $\lambda_i$'s depend on $q$, which depends on $v$, which depends on $\mu_\pi$. And at the end of Appendix D.7 is a repetition of the quote above. Update: looking at Algorithm 3 in Appendix E.3, it looks like iterations of PPO alternate with updating the lambda's. But then, a) that should be reflected better in the text, and b) it makes me question various claims that certain optimization subproblems are convex/concave as claimed. Indeed, it would be nice to know whether Algorithm 3 converges, or if it produces the kind of "chattering" that occasionally appears in deep RL.

Next, for two of four environments, results are only presented in the appendix. This looks like cherry-picking; those results should be in the main paper.

Finally, I think you should mention that Linear Max-Min has space complexity quadratic in the number of features, and time complexity cubic in it. People obviously have methods for approximating large matrices that would appear to improve the computational complexity, but these approximations inevitably obscure information about small eigenvalues, and then when you take the sqaure-root-inverse, you end up missing the entire picture, because the small eigenvalues become the most important part. It would be nice if this paper attempted to head off such mistakes should people try to extend this work to settings with thousands of features or more.

---

> ### Author Rebuttal · Authors · 2025-07-31
>
> Dear reviewer EXFr:
>
> We sincerely thank the reviewer for the thoughtful and detailed feedback. Below, we address the concerns point-by-point.
>
> Q1: Non-singularity of $Q$
>
> A1: We sincerely thank the reviewer for identifying this important issue and we agree that our initial condition is not enough for non-singularity. As correctly pointed out, for $Q$ to be non-singular, it is necessary that the span of {$ \phi (s,a):\mu_{\pi_{\text{ref}}}(s,a) >0$} covers $ R^n $, i.e., the features associated with state-action pairs visited by $\pi_{\text{ref}}$ must be linearly independent and span the full feature space. To achieve these conditions, the reference policy should visit a diverse and representative subset of the state-action space with non-trivial occupancy. This is more likely when $\pi_{\text{ref}}$ is derived from either expert demonstrations that exhibit rich behavior or from stochastic or exploratory policies (e.g., entropy-regularized policies or policies trained with exploration bonuses). Moreover, the feature mapping $\phi(s,a)$ must exhibit sufficient variation across the visited state-action pairs. This typically holds when $\phi$ encodes task-relevant dynamics (e.g., learned embeddings or expressive hand-crafted features) and when $\pi_{\text{ref}}$ does not collapse to trivial or deterministic behavior. In our experiments (Appendix E.4), the reference policies for the Traffic and Pandemic environments are trained via behavioral cloning on large, diverse trajectories generated by human experts or hand-crafted controllers. The feature representations used in these environments, such as velocity, acceleration, and headway in Traffic, and infection level, disease stage, and smooth transitions in Pandemic, encode meaningful task-relevant dynamics. These demonstrations cover a wide range of task-relevant behaviors, and the induced occupancy over state-action pairs spans a high-dimensional subspace of the feature space. We empirically verified that the resulting $Q$ matrices in our experiments are full-rank and numerically well-conditioned. We acknowledge that ensuring sufficient coverage of the feature space by the reference policy is generally challenging in practice, and we will add this as a limitation of our approach.
>
> Q2: Confusion regarding the derivation of the dual objective for Max-Min and Linear Max-Min
>
> A2:  We sincerely apologize for the confusion regarding the derivations in the Max-Min and Linear Max-Min methods, and we greatly appreciate the reviewer’s careful attention to the proofs and algorithms in the appendix. Due to space constraints in the main text, we had to place several key derivation steps (e.g., solving for the dual variables) into the appendix. We acknowledge that this disrupted the logical flow of the presentation, and we will restructure the exposition and include more essential derivations in the main text in a future revision.
>
> For the linear Max-Min algorithm, our approach is indeed an iterative process. Given a fixed policy, we first derive the worst-case linear reward. Specifically, we obtain a closed-form solution for the optimal weights $\theta$ for each feature, which depends on the dual variables $\lambda$. Unlike the Max-Min algorithm, we cannot apply the same techniques to derive a closed-form solution for $\lambda$; instead, we compute the gradient of each $\lambda$ as described in Appendix D.7 and solve for them using standard first-order optimization methods. Once the optimal $\lambda$ is obtained, we apply Equation 33 to compute the corresponding $\theta$ and construct the worst-case reward. This worst-case reward is then used to perform a PPO update on the policy. We acknowledge that this alternating procedure is not clearly reflected in the main text, and we will add a summary of Algorithm 3 to clarify this in the next version.
>
> Regarding convexity, as discussed in Appendix D.7, when $\lambda_3 < 0$, the objective in the whitened feature space becomes a convex quadratic function, ensuring that a minimum exists. We explicitly enforce $\lambda_3 < 0$ to satisfy this condition. Furthermore, since we have a closed-form solution for $\theta$ once $\lambda$ is known, for any given policy $\pi$, we effectively have access to an oracle that returns the optimal or near-optimal worst-case reward $ R^* $. Each of our algorithms 1-3 can be viewed as alternating between gradient ascent on $\pi$ and the optimal minimization on $ R^* $. As shown in Section 4 of [1], our algorithms converge, and the resulting policy $\pi$ corresponds to an approximate stationary point of the outer optimization problem.
>
>  [1] Jin, Chi, Praneeth Netrapalli, and Michael Jordan. "What is local optimality in nonconvex-nonconcave minimax optimization?." International conference on machine learning. PMLR, 2020.
>
> Q3: Complexity of Linear Max-Min
>
> We also appreciate the reviewer’s observations that computing $Q$ results in $O(d^2)$ space and $O(d^3)$ time complexity. While low-rank approximations could potentially reduce computational cost, such methods often discard small eigenvalues. However, in our setting, these small eigenvalues become critical due to the inversion in the whitening step (Equation 13) and removing them may severely distort the worst-case reward direction. We will explicitly include this discussion in our limitations to caution against naive low-rank approximations and to emphasize the need for principled, scalable extensions when applying our method to settings with very high-dimensional feature spaces.  We thank the reviewer again for highlighting this critical point.
>
> Q4: “Although the original Glucose simulator provides multiple …. feature-based form suitable for linear modeling.” What do you conclude from this? That conclusion should make it into the main paper.
>
> A4: We sincerely thank the reviewer for raising this important point. Our conclusion is as follows:
> For complex environments, constructing effective features for the reward function is often challenging without prior domain knowledge. For example, in the Glucose environment, a large number of health-related indices are provided. However, without medical expertise or knowledge of glucose monitoring, it is difficult to determine which combination of indices best captures patient health or blood glucose trends. Using arbitrarily selected features in such cases can lead to proxy rewards that exhibit little or no correlation with the true reward. While our max-min formulation can still offer robustness under such misspecification, the resulting policy is nevertheless expected to perform poorly due to the fundamental misalignment between the proxy and true objectives. Therefore, designing meaningful reward features remains a fundamental and unresolved challenge, and we will include this as a limitation of our method in the main text.
>
> Moreover, in some environments such as Tomato, the reward function is not explicitly feature-based. Although our general max-min algorithm still applies in this setting, incorporating non feature-based reward structure into the uncertainty set remains an open problem. We will include this discussion in the main text as well.
>
> Q5: On Tomato, why does Max-min have worse "worst case" reward than ORPO* when it is explicitly maximizing that? Why does Max-min enter states that have zero occupancy under the reference policy?
>
> A5: Since the Tomato environment has a discrete state-action space, we employ a sample-based estimation of the occupancy measure instead of training an auxiliary discriminator network to reduce the computational cost.  However, this estimation may be inaccurate, which can result in our Max-Min policy occasionally visiting state-action pairs not covered by the reference policy. Despite this, the estimated occupancy measure over unvisited state-action pairs for our Max-Min policy is only $1.13 \times 10^{-5}$, compared to $3.13 \times 10^{-5}$ for ORPO*, indicating that our method still achieves lower divergence from the reference policy.
>
> As state-action pairs unvisited by the reference policy can receive arbitrarily low worst-case rewards, we exclude these from the worst-case return calculation. When such pairs are included, our Max-Min policy demonstrates greater robustness than ORPO*, as it explores fewer unsupported regions of the state-action space.

---

> > ### Comment · Reviewer_EXFr · 2025-08-01
> >
> > I am satisfied with the authors' responses, and will keep my score the same.
> >
> > I'd also like to respectfully push back on some of the other reviewers' claims that the methods are not especially novel. They show convincingly that the ORPO algorithm is mistaken, and they correct it. They also show how to make ORPO less conservative in a setting where one can make assumptions about the reward functions. The concept of even being able to use assumptions to moderate ORPO is completely absent from that work. This is a much better sort of paper than a paper that makes an *unmotivated* tweak to existing algorithm, squints at a plot, and announces a new state of the art.

---

> ### Author Response · Authors · 2025-08-01
>
> We sincerely appreciate your valuable feedback and thank you for supporting our paper.

---

### Official Review · Reviewer_vpw6 · 2025-06-24

**Clarity:** 2
**Significance:** 2
**Originality:** 2
**Rating:** 4
**Confidence:** 3

**Summary:**

This paper proposes a robust optimization method for RL settings in which we only have access to a proxy reward, not the true reward.
The authors formulate this problem as a robust optimization problem over a set of reward functions with a certain correlation and show that it can be solved by maximizing the return penalized with a chi-squared divergence between the original policy and the current policy, weighted by a constant dependent on the proxy reward correlation.
They go on to consider a setting in which the robust reward set is a is limited to linear combinations of a known feature set.
Experiments in relatively simple environments show a small potential advantage of the proposed method

**Questions:**

I have two main questions:
 * From a practical point of view, it seems that if we do not know $r$ and have to treat it as a hyper-parameter (e.g. we only have a proxy reward and no knowledge about the true reward), the proposed method does not offer any advantage over ORPO? I would appreciate the author's response on this, also how $r$ was chosen in their experiments. Further, it would be interesting to see how an incorrect $r$ affects the performance of both methods.
 * How many policies were trained in each task? The appendix suggests only one policy was trained in each task, but this is not clear.

I will consider increasing my score if the concerns are addressed.

**Ethical Concerns:**

["NO or VERY MINOR ethics concerns only"]

**Final Justification:**

The rebuttal resolved my main concern (the misspecification of $r$ in practice), but the experimental validation is still rather limited.

Thus, I am increasing my score 3->4

**Limitations:**

The main limitation (as mentioned in the questions) is that we do not know $r$ in practice. It would be nice to see how the method behaves when $r$ is not known.

Another issue is that the method is very compute intensive, requiring an hour to train in a gridworld task.

**Paper Formatting Concerns:**

The tables are formatted incorrectly

**Quality:**

2

**Strengths And Weaknesses:**

Strengths:
 * The proposed method is reasonably well motivated and its derivation seems correct.
 * Applying distributional robust optimization to optimization with proxy reward models is a natural choice

Weaknesses:
 * Novelty/Utility: The resulting methods is very similar to ORPO, which the authors acknowledge. Concretely, both use chi-squared penalty with a certain weight depending on the correlation $r$.
In practice we do not know $r$, so we have to treat it as a hyperparameter to optimize.
Doing so, both methods become almost identical
 * Experimental validation is weak. It seems each method is only evaluated with a random seed and hyper-parameters are not tuned. Further the evaluated environments are relatively simplistic.
 * The paper excessively refers to the appendix and is verbose at times. A more self-contained presentation would be beneficial.

---

> ### Author Rebuttal · Authors · 2025-07-31
>
> Dear Reviewer vpw6:
>
> We thank the reviewer for the thoughtful and detailed feedback. Below we address the concerns point-by-point.
>
> Q1: How was r chosen? What is the advantage over ORPO when r is unknown? how does an incorrect r affect the performance of both methods?
>
> A1: We appreciate the reviewer’s concerns regarding the practical selection of the correlation parameter $r$ and its impact on both our method and ORPO. As discussed in Appendix E.5, the experiment results reported in the paper were obtained using the following r values for training the four environments: r = 0.3 for Traffic, r = 0.7 for Pandemic, r = 0.9 for Glucose, and r = 0.4 for Tomato. These r values were chosen following a similar approach used by ORPO. For each environment, we first performed a grid search over several different values of $r$, and for each fixed $r$, we trained one PPO policy. We then selected the $r$ value that leads to the policy with the best expected worst-case return, as shown in Tables 6 and 7 in Appendix F.2. For ORPO, we followed the ORPO paper to choose the optimal $\lambda$ (which corresponds to $\sqrt{1 - r^2} \sigma_{R_{\text{proxy}}}$). This involves evaluating a range of $\lambda$ values and selecting the one that yields the best expected return under the true reward. Note that this is infeasible in practice when the true reward is unknown during training. In contrast, our approach for choosing $r$ is more practical.
>
> Tables 6 and 7 in Appendix F.2 also illustrate the sensitivity of our approach to $r$ misspecification, where we show the worst-case performance of our policies under different training-testing $r$ value pairs. We observe that moderate values of $r$ in the range of 0.3 to 0.4 generally yield robust policies. As extremely small $r$ values tend to be overly conservative, while high $r$ values overly trust the proxy reward. Therefore, in the absence of prior knowledge of $r$, starting with a moderate $r$ is a practical heuristic.
>
> To better understand the performance of both our approach and ORPO under $r$ misspecification, Section 4.2 presents an extensive comparison under a wide range of testing $r$ values, using the fixed set of policies trained with the r values specified in Appendix E.5. Specifically, for each $r$ during testing, we sample 1,000 reward functions that satisfy the correlation constraint and report the $\textbf{average return}$ of the learned policy under these sampled rewards. As shown in Figures 1 and 5, our methods (Max-Min and Linear Max-Min) consistently outperform ORPO across nearly all values of $r$ tested, demonstrating that even when the training $r$ is misspecified, our algorithms achieve more robust and stable performance under such uncertainty.
>
> Q2: How to choose $r$ in a more principled way?
>
> A2: We acknowledge that when $r$ is unknown, both our method and ORPO lack a principled mechanism for selecting an appropriate value. Besides the simple heuristics derived from our experiments as discussed above, we outline two potential approaches to this important problem below.
>
> $\textbf{Statistical inference of r}$. If we have access to the true reward on a subset of state-action pairs, e.g., through active learning, we can estimate $r$ using Equation 2. In fact, Equation 2 defines the Pearson correlation coefficient $r$ between the true reward $R_{\text{true}}$ and the proxy reward $R_{\text{proxy}}$ under the occupancy measure $\mu_{\pi_{\text{ref}}}$. Given a batch of $n$ state-action pairs {$(s_i, a_i)$}, $i=1,...,n$, sampled from $\pi_{\text{ref}}$ for which we have both $R_{\text{true}}(s_i, a_i)$ and $R_{\text{proxy}}(s_i, a_i)$, we can estimate this correlation as follows:
>
> $\hat{r} = [\sum_i (R_{\text{true}}(i) - R^0_{\text{true}})(R_{\text{proxy}}(i) - R^0_{\text{proxy}})]/
> [\sqrt{\sum_i (R_{\text{true}}(i) - R^0_{\text{true}})^2} \cdot
> \sqrt{\sum_i (R_{\text{proxy}}(i) - R^0_{\text{proxy}})^2}]$, where $R^0$ is the sample mean.
>
> We can then use $\textbf{Fisher's z-transformation}$ to compute the confidence intervals for $r$, which can be plugged into our framework to define a tighter reward uncertainty set. For example, we can use $r_{\text{lower}}$ for more pessimistic robustness. Or we can redefine the correlation constraint in Equation 2 to be bounded by both  $r_{\text{lower}}$ and $r_{\text{upper}}$. The optimal solution under this new constraint can be similarly obtained using the approach in the paper.
>
> $\textbf{A min-max regret approach}$. A more principled approach to addressing the uncertainty in $r$ may come from a regret-based perspective. Let $J_r(\pi)$ denote the worst-case return for a given policy $\pi$ under a specific correlation level $r$, i.e., $J_r(\pi) = \min_{R \in R_{\text{corr}}(r)} J(\pi, R)$. The regret can then be defined as $Reg(\pi,r) = \max_{\pi'} $  $J_r $ ($ \pi' $) $ - J_r(\pi) $, which quantifies the performance gap between the optimal policy under $r$ and the current policy. With this formulation, a robust objective can be expressed as $\min_\pi \max_r Reg(\pi, r)$, aiming to find a policy that minimizes the worst-case regret across all possible values of $r$. This framework enables us to train policies that are robust to uncertainty in the correlation parameter $r$.
>
> Q3: Experimental validation is weak.
>
> A3: We thank the reviewer for pointing out the lack of seed variation and hyperparameter tuning in the current submission. In response, we have added new experimental results on the Traffic and Pandemic environments. Specifically, we train policies using five random seeds, consistent with the setup used in the ORPO paper. The results are summarized as follows. Due to time constraints, we are still in the process of running ORPO under five different random seeds. Nevertheless, both our methods exhibit relatively low variance, and the preliminary results remain consistent with those in the main text: our method demonstrates greater robustness while maintaining comparable performance to ORPO under the true reward.
>
> | Env            |              | Traffic     |                      |              |               |
> |----------------|--------------|-------------|----------------------|--------------|---------------|
> | Method         | True         | Proxy       | Worst                | Linear Worst | Linear Worst* |
> | Max-Min        | 12.63 ±0.05  | 3.66 ± 0.04 | -268.31 ± 4.14       | -0.06 ± 0.01 | -0.06 ± 0.01  |
> | Linear Max-Min | 16.40 ± 0.06 | 2.57 ± 0.05 | -1.19e+04 ± 0.01e+04 | 0.20 ± 0.01  | -0.12 ± 0.01  |
> | Env            |              | Pandemic      |                      |              |               |
> | Method         | True         | Proxy       | Worst                | Linear Worst | Linear Worst* |
> | Max-Min        | 1.06 ± 0.14  | 1.06 ± 0.14 | -63.29± 3.35         | -1.11 ± 0.01 | -1.11 ± 0.01  |
> | Linear Max-Min | 2.67 ± 0.11  | 7.54 ± 0.13 | -6.82e+05 ± 0.01e+05 | 0.65 ±0.01   | -0.17 ±0.02   |
>
> For the hyperparameters of training the discriminator, we adopt the overall setup from ORPO, while lightly tuning the learning rate and the number of SGD epochs to ensure stable convergence. We report the best-performing configuration in the paper in Appendix E.1, Table 2.  For other hyperparameters used during PPO training, we follow the same settings as in the ORPO paper to ensure a fair comparison.
>
> In addition to the environments presented in the main paper, we include results on an additional RLHF-style environment following the setup described in Appendix C.4 of the ORPO paper. Due to time constraints, this experiment was conducted with a single random seed. Nevertheless, consistent with our results in other environments, our method achieves improved robustness compared to ORPO, as shown below.
>
> | Env     |        | RLHF  |       |
> |---------|--------|-------|-------|
> | Method  | True   | Proxy | Worst |
> | ORPO    | 16.55  | -0.24 | -1.86 |
> | Max-Min | 16. 31 | -0.22 | -0.12 |
>
> Q4: How many policies were trained in each task?
>
> A4: We thank the reviewer for pointing out this ambiguity. For each task, we performed a grid search over several different values of $r$, and for each fixed $r$, we trained one PPO policy.
>
> Q5: The method is compute intensive.
>
> A5: We acknowledge that both ORPO and our method are computationally intensive, as shown in Appendix E.6, Table 4. This is primarily because each PPO iteration requires estimating the occupancy measure of the current policy, which in turn necessitates training an auxiliary discriminator network. While the ORPO method attempts to reduce this overhead by training the discriminator for only a few epochs (without convergence), this limits the discriminator's ability to distinguish the current policy from $\pi_{\text{ref}}$​. As a result, the robustness and stability of ORPO degrade under varying reward specifications and correlation levels, compared to ORPO* (which fully trains the discriminator), as demonstrated in Figure 1.
>
> Effectively estimating the occupancy measure is thus critical to reducing computational cost. In our work, we experimented with the following strategies. For simple environments like Gridworld, where both the state and action spaces are discrete, we avoid training a discriminator entirely. Instead, we estimate the occupancy measure using a sampling-based approach via Equation 40. This significantly reduces training time. As shown in Table 4, our method achieves a similar training time of around 1 hour as ORPO (which trains the discriminator for only one epoch), while fully training the discriminator would require approximately 2 hours. However, for more complex environments with continuous state and action spaces, this sampling-based approximation does not yield reliable occupancy estimates, so we continue to rely on training the discriminator.
>
> We acknowledge that these optimizations are insufficient to fully address the compute burden of occupancy estimation, which remains a shared limitation of both methods.

---

> > ### Comment · Reviewer_vpw6 · 2025-08-02
> >
> > I would like to thank the authors for their rebuttal.
> >
> > The rebuttal sufficiently addresses my concerns about the choice of $r$ and the experiments with miss-specfication in the appendix are indeed helpful. I would encourage the authors to mention them more prominently in the main text, as it seems quite crucial to the method. As an aside, NeurIPS allows authors to include the Appendix in the main PDF so I would recommend combining it for a camera-ready version.
> >
> > I still have reservations about the limited experimental evaluation due to lack of a structured hyper-parameter optimization, which can not easily be addressed during the time-frame of a rebuttal, so I will only raise my score 3->4.

---

> ### Author Response · Authors · 2025-08-04
>
> We thank the reviewer for the valuable feedback. Below, we provide the missing experiments for ORPO evaluated under five different random seeds. As expected, the results remain consistent with those reported in the main text: our method exhibits greater robustness while maintaining comparable performance to ORPO under the true reward.
>
> | Env            |              | Traffic     |                      |              |               |
> |----------------|--------------|-------------|----------------------|--------------|---------------|
> | Method         | True         | Proxy       | Worst                | Linear Worst | Linear Worst* |
> | ORPO        | 16.98 ±0.05  | 3.35 ± 0.05 | -1.96e+04 ± 0.02e+04       | -0.69 ± 0.01 | -0.83 ± 0.02  |
> | Max-Min        | 12.63 ±0.05  | 3.66 ± 0.04 | -268.31 ± 4.14       | -0.06 ± 0.01 | -0.06 ± 0.01  |
> | Linear Max-Min | 16.40 ± 0.06 | 2.57 ± 0.05 | -1.19e+04 ± 0.01e+04 | 0.20 ± 0.01  | -0.12 ± 0.01  |
> | Env            |              | Pandem      |                      |              |               |
> | Method         | True         | Proxy       | Worst                | Linear Worst | Linear Worst* |
> | ORPO        | -0.87 ±0.13  | 1.83 ± 0.12 | -5.31e+06 ± 0.01e+06       | -2.41 ± 0.02 | -2.65 ± 0.02  |
> | Max-Min        | 1.06 ± 0.14  | 1.06 ± 0.14 | -63.29± 3.35         | -1.11 ± 0.01 | -1.11 ± 0.01  |
> | Linear Max-Min | 2.67 ± 0.11  | 7.54 ± 0.13 | -6.82e+05 ± 0.01e+05 | 0.65 ±0.01   | -0.17 ±0.02   |
>
> We are pleased to have addressed most of the reviewer’s concerns and will include further experimental results using a structured hyperparameter optimization process in the revised paper. We sincerely thank the reviewer for the thoughtful feedback and for being willing to raise the score.

---

### Official Review · Reviewer_KVvw · 2025-06-30

**Clarity:** 3
**Significance:** 2
**Originality:** 2
**Rating:** 4
**Confidence:** 4

**Summary:**

The paper proposes a max-min (robust) reinforcement-learning framework to guard against reward hacking when agents are trained on imperfect “proxy” rewards that are only partially correlated with the true objective. Instead of regularizing against a single proxy (as in Occupancy-Regularized Policy Optimization, ORPO), the authors optimize a policy that maximizes performance under the worst-case reward drawn from the set of all proxy rewards sharing a specified correlation level with the true reward. They derive a closed-form characterization of that worst-case reward, develop practical policy-gradient algorithms, and extend the formulation to linearly-parameterized reward functions, which yields interpretable adversarial rewards. Experiments on four safety-motivated environments show that the resulting policies achieve consistently higher worst-case returns and lower performance variance than ORPO, demonstrating improved robustness and transparency in the face of reward misspecification.

**Questions:**

Can you extend your framework to this contaminated uncertainty model and still obtain analytical solutions (or tight bounds) for the inner minimization?

**Ethical Concerns:**

["NO or VERY MINOR ethics concerns only"]

**Final Justification:**

After the rebuttal and discussions with the AC and other reviewers, the authors did not fully address my main concern on the novelty of this paper. While I increase the score, I lean to reject this paper.

**Limitations:**

yes

**Quality:**

3

**Strengths And Weaknesses:**

Strengths: (1) Principled max-min formulation that directly targets reward hacking. (2) Improved interpretability through a linear variant. (3) Consistent empirical robustness across diverse tasks.

Weakness: (0) a straightforward approach by borrowing from distributionally robust RL literature. (1) not a principle way of choosing r. this is one key limitation in all these (distributionally) robust methods (2) no theoretical guarantee

---

> ### Author Rebuttal · Authors · 2025-07-31
>
> Dear Reviewer KVvw:
>
> We thank the reviewer for the thoughtful and detailed feedback. Below we address the concerns point-by-point.
>
> Q1: A straightforward approach by borrowing from distributionally robust RL literature.
>
> A1: We appreciate the reviewer’s observation that our approach is inspired by the broader distributionally robust RL. However, as discussed in Appendix C.2, our work departs from the distributionally robust RL by considering $\textbf{non-rectangular}$ reward uncertainty. In particular, we assume a correlation-constrained uncertainty set for the reward function, meaning that the adversary’s permissible deviations in reward are coupled across states. This structure can mitigate the conservativeness of the worst-case solution (the adversary cannot simultaneously push all state rewards to their extreme worst values), but it also means that the neat robustness-regularization duality from the rectangular-case no longer applies and the robust optimization must be solved (or approximated) directly. Therefore, our work tackles a form of reward uncertainty which lies beyond the scope of existing analysis.
>
> Moreover, solving the robust optimization in our setting is non-trivial. As detailed in our response to Reviewer QPmd, we encountered several key challenges. First, the mismatch in expectations, where the objective depends on the learned policy $\pi$, while the constraints are defined under $\pi_{\text{ref}}$, prevents direct optimization. To address this, we adopt a change-of-measure technique that reformulates the problem into a tractable form with consistent expectations. Second, the feasible set induced by our correlation-based constraints is non-convex due to the quadratic equality constraint on the variance, $\mathbb{E}{\mu_{\pi_{\text{ref}}}}[R^2] = V^2 + M^2$, which defines the boundary of an $L^2$ ball in the reward function space. As a result, classical convex optimization tools and strong duality are not directly applicable. Nevertheless, we provide a theoretical proof of the global optimality of our solution in Appendix D.2. Third, deriving the closed-form worst-case reward in Equation 8 requires that the learned policy avoids state-action pairs not visited by $\pi_{\text{ref}}$, since otherwise the reward can become arbitrarily negative in unsupported regions. However, as demonstrated in Section 4.2 (Table 1), this assumption does not always hold in practice. To mitigate this, we carefully train the discriminator to effectively identify such cases, as discussed in Appendix E.1.
>
> In addition to the general max-min formulation, our framework also introduces the linear structure to the reward function, which enables an interpretable characterization of the worst-case reward. This structure allows us to express the adversarial reward as a weighted combination of feature vectors, making the resulting policy behavior and the source of adversarial perturbations more transparent and interpretable. Importantly, deriving the closed-form expression in the linear case is also non-trivial; we must impose additional assumptions on the matrix $Q$ (e.g., non-singularity) to obtain analytical results, as discussed in Section 3.2 and further elaborated in our response to Reviewer EXFr.
>
> In summary, we respectfully disagree with the claim that our method is a straightforward adaptation of techniques from distributionally robust RL. Our formulation presents unique technical challenges, and our contributions, including the interpretable linear uncertainty model and its associated derivations, go beyond prior work in this area.
>
>
> Q2: Not a principle way of choosing r.
>
> A2: We acknowledge that when $r$ is unknown, both our method and ORPO lack a principled mechanism for selecting an appropriate value. We outline two potential approaches on this important problem below.
>
> $\textbf{Statistical inference of r}$. If we have access to the true reward on a subset of state-action pairs, or if such labels can be acquired through active learning, we can estimate $r$ using Equation 2. In fact, Equation 2 defines the Pearson correlation coefficient $r$ between the true reward $R_{\text{true}}$ and the proxy reward $R_{\text{proxy}}$ under the occupancy measure $\mu_{\pi_{\text{ref}}}$. Given a batch of $n$ state-action pairs {$(s_i, a_i)$}, $i=1,...,n$, sampled from $\pi_{\text{ref}}$ for which we have both $R_{\text{true}}(s_i, a_i)$ and $R_{\text{proxy}}(s_i, a_i)$, we can estimate this correlation using the sample correlation coefficient:
>
> $\hat{r} = [\sum_i (R_{\text{true}}(i) - R^0_{\text{true}})(R_{\text{proxy}}(i) - R^0_{\text{proxy}})]/
> [\sqrt{\sum_i (R_{\text{true}}(i) - R^0_{\text{true}})^2} \cdot
> \sqrt{\sum_i (R_{\text{proxy}}(i) - R^0_{\text{proxy}})^2}]$, where $R^0$ is the sample mean.
>
> We can then use $\textbf{Fisher's z-transformation}$ to compute the confidence intervals for $r$. After getting this bounded range, we can plug this bound into our framework to define a tighter reward uncertainty set. For example, we can use $r_{\text{lower}}$ for more pessimistic robustness. Or we can redefine the correlation constraint in Equation 2 to be bounded by both  $r_{\text{lower}}$ and $r_{\text{upper}}$. The optimal solution under this new constraint can be similarly obtained using the approach in the paper.
>
> $\textbf{A min-max regret approach}$. A more principled approach to addressing the uncertainty in $r$ may come from a regret-based perspective. Let $J_r(\pi)$ denote the worst-case return for a given policy $\pi$ under a specific correlation level $r$, i.e., $J_r(\pi) = \min_{R \in R_{\text{corr}}(r)} J(\pi, R)$. The regret can then be defined as $Reg(\pi,r) = \max_{\pi'} $  $J_r $ ($ \pi' $) $-J_r(\pi) $, which quantifies the performance gap between the optimal policy under $r$ and the current policy. With this formulation, a robust objective can be expressed as $\min_\pi \max_r Reg(\pi, r)$, aiming to find a policy that minimizes the worst-case regret across all possible values of $r$. This framework enables us to train policies that are robust to uncertainty in the correlation parameter $r$.
>
> Q3: No theoretical guarantee
>
> A3: We thank the reviewer for raising the concern regarding theoretical guarantees. As discussed in Appendix D.2, the inner optimization problem, i.e., finding the worst-case reward for a given policy admits a $\textbf{globally optimal closed-form solution}$ under our formulation in the tabular setting. Therefore, for any given policy $\pi$, we have access to an oracle that outputs the optimal worst-case reward $ R^* $ and each of our algorithms 1-3 can be viewed as alternating between gradient ascent on $\pi$ and the optimal minimization on $ R^* $. As shown in Section 4 of [1], our algorithms converge, and the resulting policy $\pi$ corresponds to an approximate stationary point of the outer optimization problem.
>
> [1] Jin, Chi, Praneeth Netrapalli, and Michael Jordan. "What is local optimality in nonconvex-nonconcave minimax optimization?." International conference on machine learning. PMLR, 2020.
>
> Q4: Can you extend your framework to this contaminated uncertainty model and still obtain analytical solutions (or tight bounds) for the inner minimization?
>
> A4: We thank the reviewer for raising the question about extending our framework to the contaminated uncertainty model. We would appreciate further clarification on the specific definition of the contamination model being referred to, as the term can have different interpretations in robust learning and statistics. If the reviewer is referring to the classic $\epsilon$-contamination model where the reward $R$ is modeled as: $R=(1-\epsilon)R_{\text{proxy}}+\epsilon R_{\text{adv}}$, where $R_{\text{proxy}}$ is the proxy reward, $R_{\text{adv}}$ is the arbitrary reward function representing contamination, and $\epsilon$ is the contamination level, then it is possible to adapt our framework to it. In such case, the constraint space of the rewards $\mathcal{R}{\text{corr}}$ satisfies both Equation 4 and the $\epsilon$-contamination model above. Adopting a similar derivation as in our main paper, we can get the optimal solution $ R_{\text{adv}}=\frac{L-\lambda_1 R_{\text{proxy}}-\lambda_2}{2\epsilon\lambda_3}$ for the inner minimization.

---

> > ### Comment · Reviewer_KVvw · 2025-08-07
> >
> > Thanks for addressing my comments and questions. In my opinion, correlation constraint is an example of combining the first and the second moment constraints, which is thus standard in distributionally robust optimization. Thus I don't see a significant challenge for it. For choosing r, you basically require a calibration dataset, which is fine. But under such a setting, there are many other ways of doing this. Basically it becomes a different problem.
> >
> >
> > Using minimax-regret criterion to select $r$ is an interesting idea. But why will it work?

---

> ### Author Response · Authors · 2025-08-04
>
> Dear reviewer KVvw,
>
> Thank you once again for your insightful and helpful reviews. We are eager to know if our responses have satisfactorily addressed your concerns and if you have any further questions.
>
> We would also appreciate your clarification regarding the contaminated uncertainty model mentioned in the review and would be glad to elaborate further if helpful.
>
> Thank you again for your time and consideration.

---

> ### Author Response · Authors · 2025-08-07
>
> Dear Reviewer KVvw,
>
> Thank you for your thoughtful and constructive feedback. We agree that moment-based constraints are a standard tool in distributionally robust optimization (DRO). However, we believe they remain underexplored in the context of reinforcement learning (RL).
>
> After a more thorough review, we identified only one relevant work [1] that considers uncertainty sets based on the first and second moments of the reward distribution in the RL setting. While their formulation appears similar and also uses standard KKT-based solutions, their results are not directly applicable to our max-min framework, and we still need to explicitly derive and solve our formulation, as detailed in the main paper. Furthermore, our work contributes additional important insights beyond the optimization technique itself, and the existence of this prior work does not undermine the novelty or technical contribution of the paper:
>
> 1. $\textbf{Motivation}$: Our primary motivation is to reduce reliance on proxy rewards in ORPO by explicitly modeling correlation uncertainty. The first and second moment constraints arise as a practical relaxation to make the correlation uncertainty set tractable. In contrast, [1] focuses on general reward uncertainty constraints on first and second moments.
>
> 2. $\textbf{Dependence on Reference Policy and Discriminator}$: The framework in [1] does not rely on a reference policy $\pi_{\text{ref}}$ and therefore does not require training a discriminator for occupancy estimation. In contrast, both ORPO and our method explicitly incorporate the discriminator, and we demonstrate that a well-trained discriminator is critical to achieving robustness. This connection to adversarial learning is central to our framework.
>
> 3. $\textbf{Incorporating Prior Structure}$: Our formulation allows prior structural knowledge of the reward (e.g., linearity) to be integrated into the uncertainty set, which improves both robustness and interpretability. It is unclear how ORPO can leverage such structures, and it has not been addressed in [1]. Furthermore, the framework in [1] cannot be applied to solve our linear max-min formulation.
>
> We believe these conceptual contributions are at least as important as the solution method itself, particularly from the perspective of AI safety and interpretability.
>
> We also note that under certain assumptions, the ORPO objective can be reinterpreted as a special case of the max-min formulation in [1] (Theorem 1), providing a complementary view of the connection between these approaches. Nevertheless, our optimization objective remains structurally different.
>
> Regarding the minimax-regret formulation, we think this is a promising future direction, especially for cases where $r$ may be misspecified during training.  As studied in [2], minimax-regret may provide strong robustness guarantees under distribution shifts for $r$. In such settings, methods like Prioritized Level Replay [3] and recent progress in [4] could be adapted to solve the problem by sampling multiple $r$ and solving Equation (9) in our paper. We should note that the reason these frameworks are potentially applicable is that our formulation admits a closed-form solution for the inner minimization. However, the main challenge lies in estimating the occupancy measure. An interesting direction for future work is to investigate whether policy gradients can be approximated without explicitly estimating occupancy.
>
>
> [1] Nguyen, Hoang Nam, Abdel Lisser, and Vikas Vikram Singh. "Distributionally robust chance-constrained Markov decision processes." arXiv preprint arXiv:2212.08126 (2022).
>
> [2] Sadek, Karim Abdel, et al. "Mitigating Goal Misgeneralization via Minimax Regret." Reinforcement Learning Conference.
>
> [3] Jiang, Minqi, Edward Grefenstette, and Tim Rocktäschel. "Prioritized level replay." International Conference on Machine Learning. PMLR, 2021.
>
> [4] Monette, Nathan, et al. "An Optimisation Framework for Unsupervised Environment Design." arXiv preprint arXiv:2505.20659 (2025).

---

> ### Author Response · Authors · 2025-08-08
>
> Dear Reviewer KVvw,
>
> As the discussion period draws to a close, we would like to kindly ask if our last responses have successfully addressed all of your concerns, where we further explain the contributions and insights of our paper. If so, we would greatly appreciate it if you could consider updating your final rating to reflect our discussion.
>
> Thank you once again for your time and thoughtful feedback.

---

> > ### Comment · Reviewer_KVvw · 2025-08-09
> >
> > Appreciate the response. Although I am not fully convinced by the contribution, I can increase my score because of the discussion.

---

> > > ### Author Response · Authors · 2025-08-09
> > >
> > > Dear Reviewer KVvw,
> > >
> > > We truly appreciate the constructive discussion and the opportunity to address your concerns. Thank you for being willing to raise the score.

---

### Official Review · Reviewer_QPmd · 2025-07-02

**Clarity:** 3
**Significance:** 2
**Originality:** 3
**Rating:** 3
**Confidence:** 4

**Summary:**

This paper formulates reward hacking in reinforcement learning as a robust max-min optimization problem over all proxy rewards that are r-correlated with the true reward under a reference policy. The authors derive a tractable dual form for the inner minimization, allowing efficient computation of the worst-case reward. They further extend the framework to linear reward functions over known features, resulting in a quadratic program for the adversarial case, which aids interpretability. Empirical results on several benchmarks show improved worst-case and robust performance over occupancy-regularized baselines.

**Questions:**

See weaknesses.

**Ethical Concerns:**

["NO or VERY MINOR ethics concerns only"]

**Limitations:**

A theoretical work may not pose the societal impact as a high priority.

**Quality:**

3

**Strengths And Weaknesses:**

Strengths:
I believe the robust optimization framework in reward hacking setting studied in this paper is meaningful and as the authors claimed. They derive a tractable dual form for the inner minimization in this setting and further extend the framework to linear reward functions over known features. It is also nice to see the experiment results align with the tractable prediction.

Weaknesses:
My main concern is the limited technical novelty. From the robust RL perspective, the max-min formulation considered here leads to a linear objective with respect to the reward, and the correlation-based uncertainty set imposes only linear constraints. As a result, the inner minimization is a standard linear (or quadratic, in the linear feature case) program. Thus, while the application to r-correlated proxies is interesting, the core optimization techniques and theoretical tools used in this work are relatively standard and do not represent a substantial advance over existing robust RL methods.

I'm curious about what the technical challenges are the authors met in the analysis such that the standard opitmization analysis/technical tools can not be easily applied, and how the authors overcame it/what the new techniques the authors developed?

---

> ### Author Rebuttal · Authors · 2025-07-31
>
> Dear Reviewer QPmd:
>
> We thank the reviewer for the thoughtful and detailed feedback. Below, we outline the main technical challenges we encountered during analysis and explain how we addressed them.
>
> 1) Expectation mismatch between the policy in the objective and in the constraints.
>
> As introduced in Equation 5 and discussed in Section 3.1, one major challenge arises from the fact that the expectation in the objective is taken with respect to the learned policy $\pi$, while the expectations in the constraint set $ \mathcal{R}{\text{corr}} $ are defined with respect to $\pi_{\text{ref}}$. Our initial attempt to resolve this used the projected gradient method, where we computed the gradient of the objective and projected it back onto the constraint set. However, this approach required iteratively solving both the inner and outer problems, making it significantly more computationally expensive than ORPO, which simply uses a regularization term. We ultimately found that the formulation in [1] faced a similar challenge and resolved it via a change-of-measure technique. Adopting this approach, we were able to reformulate our optimization such that both the objective and the constraints are expressed as expectations with respect to $\pi_{\text{ref}}$, leading to Equation 6.
>
> 2) Non-convex feasible set due to a quadratic equality constraint.
>
> As discussed in Appendix D.2, although the correlation-based constraint and the expectation constraint impose only linear constraints, the feasible region as a whole becomes non-convex due to the quadratic equality constraint $\mathbb{E}{\mu_{\pi_{\text{ref}}}}[R^2] = V^2 + M^2$. This constraint defines the boundary of an $L^2$ ball (i.e., a hypersphere) in the reward function space, which is non-convex. As a result, classical convex optimization tools and strong duality cannot be directly applied. To show that the resulting worst-case reward $ R^* $ in Equation 8 is still globally optimal, we proceed in two steps. First, we verify that $ R^* $ satisfies the stationarity condition. Specifically, for any fixed dual variables $\lambda_1, \lambda_2, \lambda_3$, the Lagrangian $l_0$ in Equation 7 is minimized when $\lambda_3 < 0$, and $ R^* $ is derived by setting the gradient of $l_0$ to zero. This confirms that $ R^* $ satisfies the first-order stationarity condition and lies in the domain where the Lagrangian is well-defined and differentiable. Second, we verify the feasibility of the closed-form primal solution $ R^* $($ \lambda^* $), where $ \lambda^* $ denotes the optimal dual variables. Specifically, we check that the optimal values of $ \lambda_1^* $, $ \lambda_2^* $, and $ \lambda_3^* $, derived in Appendix D.1, satisfy the three equality constraints in the feasible set $\mathcal{R}{\text{corr}}$. Therefore, the solution $ R^* $($ \lambda^* $) is feasible. Since $ R^* $ satisfies both stationarity and feasibility, we conclude that it is a globally optimal solution to the inner minimization problem.
>
> 3) Worst-case reward degeneracy when $\mu_{\pi_{\text{ref}}}(s,a) = 0$.
>
> To derive the closed-form expression for the worst-case reward in Equation 8, it is necessary that $\mu_{\pi_{\text{ref}}}(s,a) > 0$ for all $(s,a)$; otherwise, the adversary can assign arbitrarily negative rewards in those regions. While the ORPO paper similarly assumes that the learned policy avoids state-action pairs not visited by $\pi_{\text{ref}}$, our experiments (Section 4.2, Table 1, Appendix F.2, Table 5) show that policies trained with ORPO may still explore such regions. To mitigate this, we propose detecting these state-action pairs and assigning large negative rewards to them. Initially, as discussed in Section 3.3, we attempted to identify such regions using the trained discriminator. However, we observed that the discriminator is ineffective at distinguishing between $\pi_{\text{ref}}$ and $\pi$. After several experiments, we found in the original ORPO implementation, the discriminator was often under-trained. We addressed this by carefully tuning the discriminator’s learning rate and training epochs to ensure convergence, as detailed in Appendix E.1. Experimental results in Section 4.2 and Appendix F. 2 show that ORPO*, which fully trains the discriminator, achieves a lower occupancy measure over state-action pairs unseen by $\pi_{\text{ref}}$ compared to ORPO. These confirm that, with proper tuning, the discriminator can more effectively discourage exploration into unsupported regions of the state-action space, thereby improving the robustness and stability of the learned policy across varying $r$, as illustrated in Figure 1.
>
> 4) Deriving the closed-form solution under linear reward assumptions.
>
> The optimization problem in Equation 5 becomes significantly more complex when incorporating prior structure into the reward function. As linearity is one of the simplest and most interpretable assumptions, we begin by exploring this setting. As discussed in Section 3.2, the problem in Equation 12 does not admit a universal closed-form solution under arbitrary $Q$. To enable analytical progress, we impose additional assumptions on $Q$. Specifically, $Q$ is positive semi-definite and it can be non-singular under the assumptions that the $\pi_{\text{ref}}$ should explore a large subset of the state-action space and those visited state-actions should activate the full feature space. Under these conditions, we can apply a whitening transformation using Cholesky decomposition (Appendix D.5), which simplifies the problem to Equation 13, where a closed-form solution for the worst-case reward can be derived. However, even under this simplified setting, we are unable to derive a closed-form solution for the associated dual variables $\lambda$. Instead, we rely on standard first-order optimization solvers to compute the optimal values of these variables. While this formulation is still solvable, it is substantially more complex than the original closed-form expression derived for the reward without any prior structure knowledge. For reward models with even more general structures, such as parameterizing the reward function via a neural network, the inner minimization problem becomes even more intractable. As acknowledged in Appendix A, such cases do not admit closed-form solutions, and iterative adversarial training between the policy and the reward becomes necessary, unlike the standard and linear settings. This highlights a fundamental trade-off between the complexity of the reward structure and analytical tractability in the current framework.
>
> [1] Hu, Zhaolin, and L. Jeff Hong. "Kullback-Leibler divergence constrained distributionally robust optimization." Available at Optimization Online 1.2 (2013): 9.

---

> > ### Comment · Reviewer_QPmd · 2025-08-06
> >
> > I really appreciate the response by the authors, which addresses some of my questions/concerns.
> >
> > However, I'm still not convinced that the technical novelty is enough for acceptance. I tend to keep my score.

---

> ### Author Response · Authors · 2025-08-05
>
> Dear Reviewer QPmd:
>
> Thank you once again for your insightful and helpful reviews. We are eager to know if our responses have satisfactorily addressed your concerns and if you have any further questions.
>
> Thank you again for your time and consideration.

---

> ### Author Response · Authors · 2025-08-06
>
> Dear Reviewer QPmd:
>
> We thank the reviewer for the feedback and are pleased that some concerns have been addressed. We would also appreciate further clarification on any remaining issues.
>
> As detailed in our rebuttal, we respectfully clarify a potential misunderstanding: while our formulation for the max-min case may appear linear at first glance, the inner problem is in fact quadratic, involving a non-convex constraint due to the fixed variance condition. Consequently, standard robust RL techniques are not directly applicable to our setting.
>
> To our understanding, technical contributions of a paper are not just about improving existing algorithms or proposing new ones, but also providing fundamental new insights into a research problem. In our paper, we identified some important weaknesses of ORPO, including its reliance on under-trained discriminators and the impractical way of choosing $r$. More importantly, we demonstrate the importance of exploiting the structure of reward functions to improve both robustness and interpretability, whereas it is unclear how ORPO could leverage such structure to reduce conservativeness. These findings and insights are equally important, if not more important, than adapting the robust RL framework to an AI safety problem.
>
> We believe that our paper provides substantial technical contributions in both theory and methodology, and we greatly value a more concrete discussion with the reviewer to better understand your remaining concerns and how we might further address them.

---

> ### Author Response · Authors · 2025-08-08
>
> Dear Reviewer QPmd,
>
> As the discussion period draws to a close, we would like to kindly ask if our last responses have successfully addressed all of your concerns, where we further explain the contributions and insights of our paper. If so, we would greatly appreciate it if you could consider updating your final rating to reflect our discussion.
>
> Thank you once again for your time and thoughtful feedback.

---

### Note · Authors · 2025-08-11

Dear Reviewers and AC,

Thank you once again for your time and effort in reviewing our paper and for providing insightful, constructive feedback. We would like to take this opportunity to clarify our contributions again and formally conclude the author-reviewer discussion.

A common concern among reviewers has been the technical novelty of the proposed max-min framework, which is standard in robust optimization. However, we believe it is underexplored in the context of reinforcement learning, and existing robust optimization techniques cannot be directly applied to our problem without non-trivial adaptations and rigorous proofs.  Furthermore, as highlighted by Reviewer EXFr, our work makes important contributions beyond the optimization technique itself:

1. $\textbf{Well-trained discriminator: }$ We demonstrate that a well-trained discriminator is critical for achieving robustness, a point that has been under-explored in ORPO.

2. $\textbf{Incorporating Prior Structure: }$ Our formulation allows prior structure knowledge of the reward (e.g., linearity) to be integrated into the uncertainty set, which improves both robustness and interpretability.  It is unclear how ORPO can leverage such structures to reduce conservativeness.

We believe these contributions offer novel insights into AI safety and interpretability, while also bringing fresh perspectives to the robust RL literature.

We would also want to thank all reviewers for their constructive feedback and thoughtful engagement, including their input on selecting $r$, strengthening robust evaluation, and connecting our work to the distributionally robust optimization literature. In particular, we thank Reviewer EXFr for recognizing our technical contributions and for offering valuable suggestions that have helped improve the completeness and correctness of our work. We are pleased to have addressed most concerns and appreciate the reviewers’ decisions to raise their scores. These discussions have not only strengthened our paper but also helped us better articulate our contributions.

Sincerely,

Authors of Submission 25047

---

### Decision · Program_Chairs · 2025-09-17

**Decision:**

Reject

**Comment:**

The paper considers the RL setting with a proxy reward but not a true reward, and reward hacking may often occur in this scenario. To improve worst-case performance, the authors considered a robust optimization formulation over a set of reward functions sharing a specified correlation level with the true reward. Theoretical analysis was conducted to characterize the worst-case reward, and the interpretability and robustness were analyzed for the tractable case where rewards are linearly parameterized. A practical implementation of the proposed algorithm generally shows a potential advantage of the proposed method in a few representative environments.

The reviews of this paper are mixed, and one common concern raised by reviewers was that the paper is a combination of robust optimization and an existing algorithm, ORPO. In addition, the contribution of this paper can definitely also be improved with more comprehensive numerical results.

Since the paper is on the borderline, the AC has discussed with the SAC for the final decision on this paper. After discussion, we believe that although the minimax formulation here is sufficiently interesting, the author did not make clear the contribution of this paper in the context of earlier work in pessimism RL formulations, where the minimax formulation here is well known. Chi-2 regularization (together with minimax formulation) has been explored earlier theoretically in https://arxiv.org/pdf/2202.04634, and more recently in the RLHF context by https://arxiv.org/pdf/2407.13399. Although the paper has a different angle, these related works should be cited and discussed to better clarify the significance of this work.

We hope the authors will strengthen this paper for the next venue following the comments from the reviewers and the ACs.